# Healthcare Professionals’ Views and Perspectives towards Aging

**DOI:** 10.3390/ijerph192315870

**Published:** 2022-11-29

**Authors:** Peggy Palsgaard, Christian A. Maino Vieytes, Natasha Peterson, Sarah L. Francis, Lillie Monroe-Lord, Nadine R. Sahyoun, Melissa Ventura-Marra, Lee Weidauer, Furong Xu, Anna E. Arthur

**Affiliations:** 1Carle Illinois College of Medicine, The University of Illinois at Urbana-Champaign, Urbana, IL 61801, USA; 2Division of Nutritional Sciences, The University of Illinois at Urbana-Champaign, Urbana, IL 61801, USA; 3Department of Human Development and Family Studies, The Iowa State University, Ames, IA 50011, USA; 4Department of Food Science and Human Nutrition, The Iowa State University, Ames, IA 50011, USA; 5Center for Nutrition, Diet, and Health, The University of the District of Columbia, Washington, DC 20008, USA; 6Department of Nutrition and Food Science, The University of Maryland, College Park, MD 20742, USA; 7Department of Nutritional Sciences, West Virginia University, Morgantown, WV 26506, USA; 8School of Health and Consumer Sciences, South Dakota State University, Brookings, SD 57007, USA; 9School of Education, University of Rhode Island, Kingston, RI 02881, USA; 10Department of Dietetics and Nutrition, The University of Kansas Medical Center, Kansas City, KS 66160, USA

**Keywords:** ageism, anxiety, education, geriatrics

## Abstract

Improving care for the older population is a growing clinical need in the United States. Ageism and other attitudes of healthcare professionals can negatively impact care for older adults. This study investigated healthcare professionals’ (*N* = 140) views towards aging and characterized a confluence of factors influencing ageism perspectives in healthcare workers using path analysis models. These models proposed relationships between aging anxiety, expectations regarding aging, age, ageism, and knowledge. Aging anxiety had a less critical role in the final model than hypothesized and influenced ageism in healthcare workers through its negative effect (β = −0.27) on expectations regarding aging. In contrast, aging knowledge (β = −0.23), age (β = −0.27), and expectations regarding aging (β = −0.48) directly and inversely influenced ageism. Increased knowledge about the aging process could lower ageism amongst healthcare professionals and improve care for older adults. The results put forth in this study help to characterize and understand healthcare workers’ complex views towards the aging population they often encounter. Moreover, these results highlight the need and utility of leveraging practitioner education for combating ageism in the clinical setting.

## 1. Introduction

Over the next forty years in the United States, there will be an unprecedented increase in the older adult population. By 2030, 71 million adults are expected to be 65 or older. By 2060, this number will increase to 98 million, making up approximately 25% of the population [1]. Older adults are more likely to utilize healthcare systems and present with multiple conditions [2]. This creates a need to ensure quality healthcare for the older population [3]. Ageism, which encompasses an individual’s perception of older adults as well as actions towards them, has been shown to lead to negative health outcomes of older patients on a systemic scale [4,5]. As an example, a 2014 systematic review found that older adults were underrepresented in clinical trials relating to lower back pain [6]. Further, the impact of individual ageist views has been shown to negatively affect health outcomes of older adults and even to have a significant economic cost [7,8,9]. Ageist views among healthcare workers, their knowledge about the aging process, and attitudes influence the quality of care they provide older patients [10]. Ageism amongst healthcare workers is a problem that directly affects patient care.

There is a complex relationship between aging anxiety, aging knowledge, expectations regarding aging, and ageism. In the general population, there has been research identifying associations between aging knowledge and aging anxiety [11]. Barnett and Adams showed an association between aging knowledge and lower ageism while aging anxiety was associated with greater fear of death [12]. Allan and Johnson reported aging knowledge and contact with the older adult population indirectly affected ageism, mediated by aging anxiety [13]. Knowledge and contact with the older population were directly associated with aging anxiety, and aging anxiety was directly associated with ageism [13].

While numerous studies have explored the relationships between aging knowledge, ageism, and aging anxiety, there is limited research examining these relationships within the healthcare worker population. Healthcare workers are a unique population in that they are often a key source of health information for older patients [14,15]. A healthcare worker’s views towards their older patients can alter the care and relationship they have with the patient [10]. Ageism is present at both a systemic level in healthcare and on an individual level [4]. A systematic review from 1996 argues that individual healthcare workers should address their negative attitudes towards older adults in order to change the “nature and shape” of programs for older adults [16]. A more recent study from 2016 supports this recommendation as it found increased exposure to age stereotypes and discrimination in the healthcare system may increase the risk of chronic disease, mortality, and other adverse health outcomes [17]. Among healthcare workers, aging anxiety is positively associated with job satisfaction and career commitment to working with older adults; knowledge and attitudes towards aging of nurses are associated with varying patient interactions in emergency department settings. [10,18]. The relationship between all the factors of ageism has not yet been elucidated. While ageism is present on a systemic level in the healthcare field, characterizing and understanding the complex factors affecting ageism can positively impact older patients’ lives and outcomes.

The downstream ramifications of ageism in healthcare are important to consider in clinical contexts. A recent systematic review and meta-analysis of 422 studies assessing the impacts of ageism on measurable health outcomes found a significant deleterious association between ageism and a broad range of health outcomes examined [5]. In particular, this analysis found that ageism was associated with significantly worse health outcomes across eleven domains of health [5]. A potential explanation of these phenomena involves the accelerating demand and time pressures placed on healthcare workers, which result in stereotyping patients and biased medical decision-making [19]. Concomitantly, there is extensive documentation in the literature delineating the ways by which ageism leads to restricting older person’s access to healthcare services [5]. Limits to access have not been exclusive to services but also to clinical trials involving potentially life-saving treatment, as previously noted [20]. Thus, there is growing concern about ageism amongst healthcare workers and the impact it has on patient outcomes [4]. The COVID-19 pandemic also highlighted the negative effects of ageism on the older adult population [21]. A recent systematic review found that attitudes among nurses towards older people are complex, contradictory, and warrant further exploration [22]. Further research is warranted into the factors affecting ageism to improve patient outcomes.

Donizzetti showed that aging anxiety and aging knowledge was associated with increased aging stereotypes, which then influenced ageism among the Italian population; however, this model suggested effect modification by gender [23]. This model also did not include a link between age and aging knowledge, which could be a relevant connection as knowledge of the aging process gradually can increase as one ages [23]. While this model has limitations, it was used to guide this study’s hypothesized model as it is the most recent and relevant model according to these authors’ literature review. The goal of this analysis was to deepen our understanding of the views harbored by healthcare workers on aging. Accordingly, a path analysis model was presented utilizing a cross-sectional survey of U.S. healthcare workers, which explores the relationship between age, ageism, aging anxiety, expectations regarding aging, and knowledge about the aging process.

## 2. Materials and Methods

### 2.1. Setting

This was a cross-sectional survey of a convenience sample of 140 healthcare workers participating in a national training needs and preference assessment that was conducted online. Participants were recruited by the Qualtrics™ research team via email invitation or prompted on the respective survey platform to proceed with the survey (Qualtrics, Provo, UT, USA). The survey invitation provided information on the incentive offered and provided a hyperlink that took the respondent to the survey. To avoid self-selection bias, the survey invitation did not include specific details about the contents of the survey. The online survey was comprised of 96 questions. The questions inquired about the participant’s sociodemographic factors, healthcare experience, age-related issues (i.e., knowledge, anxiety, ageism), gerontology training needs, and education style preference.

### 2.2. Participants and Procedures

The inclusion criteria for the participants were as follows: (1) above 18 years old; (2) resident of the US; (3) healthcare worker; (4) able to read and understand the survey questions (5) have internet access; and (6) serving on one of the market research panels contracted with Qualtrics™. Qualtrics™ survey responses were then transferred to R for data analysis. Participants received an incentive based on the length of their survey, their specific panelist profile, and target acquisition difficulty. Rewards varied and included cash, airline miles, gift cards, redeemable points, sweepstakes entrance and vouchers (up to a USD 4.20 value).

### 2.3. Exclusions

Our initial dataset included 174 participants. A total of 34 respondents were removed for this analysis as 29 failed the Qualtrics™ quality check, and five did not have complete responses for aging anxiety, expectations regarding aging, ageism, and knowledge of aging, preventing tabulation of a total score. The Qualtrics™ quality check is a provided screening tool that takes into account variables such as repeat responders, possible bots, and respondents that finished the survey abnormally fast [24]. Thus, the final sample size was 140 participants.

### 2.4. Instruments

The instruments utilized in this study to measure the primary variables of interest were the Expectations Regarding Aging questionnaire, the WHO ageism scale, the Aging Anxiety Scale, and the Facts on Aging Quiz [25,26,27,28]. These instruments are described in more detail below.

### 2.5. Expectations Regarding Aging

The Expectations Regarding Aging questionnaire is a previously validated tool which examines three general areas: (1) Expectations Regarding Aging Physical Health, (2) Expectations Regarding Mental Health, and (3) Expectations Regarding Cognitive Function [25]. Each has four questions, for a total of 12 responses. The questionnaire asks about various components of aging, such as “It’s normal to be depressed when you are old,” to which participants respond “definitively true, somewhat true, somewhat false, or definitively false.” These answer options are given a score of 1–4, respectively. For each of the three subsets, the scores were totaled, subtracted by 4, and multiplied by 6.25. This yields a score within the 0–100 range. Higher scores indicate higher expectations, while lower scores indicate lower expectations [25]. In this sense, higher scores are related to elevated expectations on achievement and maintenance of physical and mental acuities throughout the aging process [25].

### 2.6. World Health Organization (WHO) Ageism

The WHO has a simple ageism quiz available, with eight *True* or *False* statements such as “poor health is inevitable in older age.” False statements are assigned a “1” while true statements are assigned a “0”, and the eight statements were totaled for a numerical score of 0–8. A higher score reflects a more ageist view [26].

### 2.7. Aging Anxiety Scale

Lasher and Faulkender’s validated Anxiety about Aging (AAS) scale was utilized in this study [28]. The scale contains 20 items and has four subscales which address: (1) Fear of Older Adults, (2) Psychological Concerns, (3) Physical Appearance, (4) Fear of Loss [28]. The participants were given statements such as “I expect to feel good about life when I am old” and responded on a Likert-type scale ranging from 1 (strongly disagree) to 5 (strongly agree). Higher scores indicate a greater anxiety regarding aging. Each subset had a total score of 25, so when added altogether the score has a max of 100. In this study, the four subscales were totaled to yield a general anxiety score, as done in previous work [29].

### 2.8. Facts on Aging Quiz

The Facts on Aging quiz is a measure of knowledge regarding aging, which includes 50 items with response options of *True* and *False.* The quiz includes true or false questions which asked about the health of older persons, including “Memory loss is a normal part of aging.” The Palmore Aging Quiz has been available and used since 1977; however, this study utilized a revised and validated version from 2015 [27,30]. It includes approximately half the original Palmore questions, as well as updated information to be of significant interest currently [27]. The test is scored by adding the number of correct answers, with scores ranging from 0 to 50. A higher score reflects a more accurate knowledge of aging.

### 2.9. Statistical Analysis

Descriptive statistics were tabulated to evaluate participant characteristics. To assess relationships amongst the measured scales, a Spearman correlation coefficients matrix was computed. Cronbach’s alpha was used to assess the internal validity of the three scales used in the analysis. We used a standard of *α* > 0.60 to deem the scales reliable and acceptable for the analysis [31]. The aging anxiety scale (AAS) consisted of twenty items (*α* = 0.72), the expectations regarding aging (ERA-12) scale included 12 items (*α* = 0.90) with each 4-item subscale demonstrating high internal consistency (*α*’s ranging from 0.71–0.82), the knowledge scale contained 50 items (*α* = 0.60), and the ageism scale comprised eight items (*α* = 0.72).

Ageism is a complex phenomenon linked to many factors [11,12]; therefore, path analysis was utilized due to its ability to adequately examine relationships amongst multiple variables with the fitting of simultaneous regression equations and its ability to estimate both direct and indirect relationships [32]. This approach was carried out by fitting five models. First, we fit the model corresponding to the hypothesized conceptual framework (Figure 1). This model hypothesized direct relationships between age on ageism, expectations regarding aging on ageism, knowledge on expectations regarding aging, age on aging anxiety, knowledge on aging anxiety, age on expectations regarding aging, aging anxiety on ageism, and aging anxiety on expectations regarding aging. It also included indirect effects of age on ageism mediated through expectations regarding aging and aging anxiety on ageism mediated through expectations regarding aging. Second, in model 2 we added a correlation between age and knowledge, given previous reports on the relationship between age and aging knowledge and the fact that the conceptual model did not account for this [31]. In model 3, non-significant paths were trimmed from the model before adding a direct path from knowledge to ageism in model 4 [12,13]. Finally, in model 5, we added an indirect effect of knowledge on ageism mediated through expectations regarding aging to assess its significance. Several measures of goodness-of-fit and their corresponding thresholds were employed as described in the SEM literature to evaluate both the measurement and structural models. These included the Normed Fit Index (NFI) and Comparative Fit Index (CFI), the Root Mean Square Error of Approximation (RMSEA) < 0.06, Standard Root Mean Square Residual (SRMR) < 0.08, $\frac{\chi^{2}}{df}$ < 3, and $\chi^{2}$ tests to evaluate model fit and to test differences between nested models [33,34]. Given that the $\chi^{2}$ test and the RMSEA are sensitive to sample size and degrees of freedom, respectively, we did not rely primarily on these fit metrics for our assessment of model fit [34,35,36]. Estimation of all measurement and structural models proceeded with robust maximum likelihood estimation (MLR) [37,38]. We report only standardized path coefficients and factor loadings.

We investigated for any moderating effect of gender on the final structural model given that gender was previously demonstrated to modify effects in the investigated relationships in an analysis using similar scales [23]. For the moderation analysis, we fit the final path model on each level of gender (“male” and “female”) and first allowed the path coefficients to vary freely across groups. A second model was fit whereby path coefficients and factor loadings were constrained to be equal across the two groups, and the Satorra–Bentler scaled $\chi^{2}$ difference goodness-of-fit test was used to assess whether the model without parameter constraints was a better fit compared to the model with constraints [39,40,41]. All analyses were conducted at *α* = 0.05 and done in R v4.2.1 (The R Foundation, Vienna, Austria).

## 3. Results

### 3.1. Descriptive Analysis

Table 1 demonstrates the demographic information of our sample. The sample was largely not Spanish, Hispanic, or Latino, and the most predominant race was white (78.6%). A majority of subjects reported having at least some college or greater and a substantial majority of the sample comprised female participants.

Table 2 provides information on the scores used in the final models. Many averages were approximately at the midline. For example, the aging knowledge quiz and ageism both had averages of approximately 50% their total range.

### 3.2. Correlation Analysis

Bivariate relationships were examined with Spearman correlation coefficients (Table 3). Ageism was moderately and negatively correlated with each subdomain of the ERA scale and the composite sum of those subdomains. Similarly, greater knowledge about the aging process was negatively correlated with ageism. There was a more modest correlation between age and ageism, which was again in the negative direction. However, there was a weak relationship between AAS scores and ageism.

### 3.3. Measurement and Path Analysis Models

We utilized a previously tested model as the initial conceptual model (Figure 1) [23]. For the aging anxiety and expectations regarding aging constructs, we utilized the composites by taking the sum of each scale’s subdomains. Some of the AAS subdomains yielded low magnitudes of Cronbach’s *α* (*α*’s ranging from 0.41–0.85). Thus, we made a decision to have the total summed scores (*α* = 0.72) measure the aging anxiety construct as previously done [29]. In this study, we are more interested in an individual’s overall aging anxiety rather than individual fears on appearance or loss, especially since we are targeting healthcare workers who have unique experiences compared to the general population. We felt that excluding those with a low Cronbach *α* would skew the data more than including them since the overall validity was good. Moreover, fitting the model in this manner allowed us to circumvent any issues with model identification, always yielding over-identified models. Consequently, we proceeded with fitting the conceptual model. The statistics reported in Table 4 demonstrate that the model fit indices were fair for this model. Adding a path from age to knowledge did not significantly alter the goodness-of-fit metrics. Nevertheless, there were several non-significant paths that were removed. Following the addition of a path from knowledge onto ageism, we arrived at the final model, which retained only significant paths and demonstrated a substantially better fit to the data relative to all others fit in the model building process. Relative to the hypothesized conceptual model, all model fit indices improved, and the $\chi^{2}$ test was no longer significant, allowing us to conclude that model specification was adequate. All goodness-of-fit metrics exceeded their a priori-defined thresholds defined above.

A summary of direct and indirect effects from both the hypothesized and final models is summarized in Table 5 and Figure 2 diagrams the paths and standardized coefficients (β) for Model 5. All coefficients from direct and indirect paths in the final model were significant at the α = 0.05 level.

The final model showed that greater aging knowledge (β = −0.23) and increased age of the health professional (β = −0.27) were associated with diminished ageism. Similarly, holding higher expectations regarding aging (β = −0.48) was also associated directly with lower ageism while also mediating indirect effects of aging anxiety and knowledge on ageism. That is, greater aging anxiety negatively impacted one’s expectations regarding aging (β = −0.27), which in turn was associated with a weak increase in ageism (β = 0.13). Notably, the direct effect of aging anxiety on ageism tested in the conceptual model produced a coefficient that was negligible and non-significant resulting in it being dropped from the final model. Considering these final results substantiates the notion that aging anxiety precedes expectations regarding aging and subsequently impacts ageism. Finally, greater knowledge of the aging process also influenced ageism directly (β = −0.23) and indirectly (β = −0.15) by increasing one’s expectations regarding aging (β = −0.32), in both instances attenuating ageist attitudes.

In the moderation analysis, we found that conducting the analysis separately in males and females did not produce significantly different results that would signal the stratified data fit the models differently ($\chi_{Satorra-Bentler}^{2}$ = 3.34, *p* = 0.134).

## 4. Discussion

A growing older adult population has spurred research that aims to identify factors that affect ageism, expectations regarding aging, aging anxiety, and aging knowledge with the goal of improving care for the older population. Previous studies of healthcare workers have identified relationships between individual factors presented in this model, while similar models have been presented in other populations but not the healthcare workforce. Our results revealed aging knowledge was directly related to expectations regarding aging; however, aging anxiety was not directly related to knowledge. The healthcare worker’s age had a negative effect on ageism, which interestingly conflicts with our conceptual model. Age was not directly related to expectations on aging or anxiety. Aging anxiety negatively predicted expectations regarding aging, which indirectly predicted ageism. Finally, an indirect effect of aging anxiety on ageism was seen with expectations regarding aging as a mediator.

Aging anxiety has been a heavily studied factor in identifying both the general population and healthcare workers’ attitudes towards aging. It has further been suggested that a higher prevalence of aging anxiety would be expected in healthcare workers, as this population is exposed to the most ill older adults requiring frequent medical interventions [4]. Thus, in our conceptual model, aging anxiety had a direct inverse relationship with expectations regarding aging and ageism. We predicted that knowledge would indirectly affect ageism through its effects on aging anxiety and expectations. However, in our final model, aging anxiety only directly affected expectations regarding aging, which in turn indirectly affected ageism. Aging anxiety was not related to knowledge in this model. This is particularly interesting in the context of existing research in the general population, which found that knowledge with older adults affects ageism, but indirectly through an effect on anxiety [13]. In the study conducted by Allan and Johnson in 2008, aging anxiety affected ageism indirectly and was mediated by expectations regarding aging rather than knowledge [13]. In our own study, anxiety was weakly and negatively correlated with knowledge (Table 3). Our results suggests that when other variables are considered, anxiety regarding one’s own aging plays a less significant role in predicting ageism than knowledge, age, and expectations regarding aging do. This finding could differ from the hypothesized model due to the unique relationship healthcare workers have with knowledge and anxiety regarding aging and could support the need to study healthcare workers specifically when addressing anxiety regarding aging.

Knowledge proves to be an important factor in our model from a practical standpoint as it is a variable that can be improved upon on an individual basis. It directly affects ageism, although it is the weakest predictor amongst age, knowledge, and expectations. Improving knowledge on the aging process relates to higher expectations regarding aging, which decreases ageism. Further, increasing knowledge directly relates to a decrease in ageism. This finding suggests that a successful strategy for reducing ageism in healthcare is to improve healthcare workers’ knowledge about the aging process.

Age only affected ageism directly in the final model. Older age was related to less ageism in the study population. This was a predictable effect from the correlation results as age was most related to ageism. Regardless, this is in contrast with the conceptual model, in which age had direct effects on expectations, anxiety, and ageism. The decreased effect of an individual’s age is an encouraging finding, as age is not a variable that can be improved or changed. Despite the inflexibility of the variable, how an individual’s age can affect their views is an important factor to recognize.

There are multiple deviations in our findings from our hypothesized model outlined above. The hypothesized model was built largely off the work of Donizzetti, which differed from our study in multiple ways. Donizzetti found that there were differences when the model was fit by level of sex while our model did not [23]. This study relied on a smaller sample size of U.S. healthcare workers, while the work of Donizzetti was done with a larger sample size in Italy and was not specific to healthcare workers [23]. Interestingly, there have been conflicting reports in the literature concerning ageism and the moderating effect of gender or sex. While we were not able to ascertain this phenomenon in these data, similar to Donizzetti, others have reported more positive attitudes towards aging in female healthcare workers [42]. Concomitantly, other studies have reported, similar to ours, no moderating effect of sex or gender on ageism [43,44,45]. Nevertheless, we also point to the fact that the majority of our respondents identified as women, which likely left us underpowered for studying this relationship with greater granularity. Considering the mixed results, we propose that further research is needed to understand the relationship that sex or gender have on ageist attitudes amongst healthcare workers. Finally, age has a varying significance in the predicted and final model, which could be due to the difference in average age, 35.8 and 40.2 for Donizzetti and this study, respectively [23].

This study has multiple strengths and weaknesses. Our sample included a large percentage of female participants. However, according to census data from 2013–2019, women hold approximately 76% of all healthcare jobs [46]. Therefore, the sample presented here (77.9% women) could be considered representative of the healthcare field. Further, this model considers healthcare workers rather than the general population. As healthcare workers have different exposures to the process of aging than the general population, this study is uniquely suited to give insight which can be utilized to limit healthcare workers’ ageist biases. Despite that our study intended to reach all healthcare workers, the majority of participants were nurses, which may limit the generalizability of the findings to other healthcare workers. Another notable weakness included the limited sample size for path analysis. A commonly cited minimum sample size figure for structural equation modeling and path analysis is around 200 samples [31]. However, more recent analysis has shown that a wide range of sample sizes can be used for these analyses and, despite this limitation, we observed excellent model fit statistics [47]. Nevertheless, we still contend that path analysis is the best approach for modeling these data given the ability to model complex relationships amongst several variables, which was the premise of our analysis. Furthermore, the WHO ageism tool utilized in this study has not yet been validated in the healthcare population. All other scales included in this study have been previously validated. Finally, we note that given the observational study design, residual confounding cannot be ruled out.

The results we present provide a framework for understanding how a healthcare worker’s expectations, anxiety, age, and aging knowledge relate to each other and influence ageism in healthcare workers. As knowledge is a variable that can be improved through trainings, workshops, and required education, it is an encouraging finding that increased knowledge improves expectations and lowers ageism. Age plays a less important role in the model than hypothesized, which is another encouraging finding as age cannot be changed. Aging anxiety had less significance than hypothesized; however, it still affects expectations on aging and ageism indirectly. Future directions of this work could include repeating this model with a larger and more diverse sample and studying the effects of improving knowledge on healthcare workers’ ageism toward their patients. By understanding how healthcare workers view and interact with their older patients, we can take steps towards improving care for the growing older population in the United States. Approaches for mitigating ageism in healthcare may include developing pilot training programs for healthcare workers that target ageist attitudes and can be validated in pilot and, subsequently, randomized controlled trials.

Nevertheless, policy considerations are also justified to try and address ageism in light of studies, including ours, that underscore the relevance of ageism as a social determinant of health with documented effects on the health of those subjected to it. Firstly, though policies involving laws and measures that protect human rights and the rights of those in the aging population are necessary, we posit that our findings justify the development of policies governed at the level of medical institutions and organizations for developing curricula that train healthcare workers on aging misconceptions, stereotypes, and the financial, health, and sociological ramifications of ageism in clinical contexts [48]. This approach requires medical organizations and public health agencies recognize and elevate ageism as a social determinant of health that mandates action. Secondly, systems for sound data collection on ageism administered by these organizations are required for further research and action. Ageism is a complex phenomenon that is challenging to measure [48]. Public and private initiatives will be necessary for fostering a movement that strives to implement data collection systems to measure and subsequently mitigate ageism amongst the healthcare workforce.

## 5. Conclusions

In conclusion, our findings highlight the complex relationships that underlie ageism, either consciously or subconsciously, in healthcare workers. Aging knowledge, higher expectations regarding aging, and increasing age were associated with lower ageism amongst healthcare workers. Greater aging anxiety was associated with lower expectations of the aging process, which, in turn, predicted a more ageist concept. Nevertheless, we call to attention and echo previous calls for a validated and reliable scale that comprehensively measures the multidimensional nature of ageism [49,50]. Despite the other limitations of our study, primarily relating to sample size, we were able to underscore important relationships between factors implicated in ageism. These relationships are meant to encourage policy changes at the level of healthcare provider education. Moreover, our findings justify a discussion on the need for a shift in the framing of aging in healthcare provider education and in broader academic instruction. Moralizing paradigms that emphasize the negative facets of aging, particularly focusing on the incidence and prevalence of age-related diseases, are ubiquitous in these contexts [51]. Consequently, combatting ageism demands a more nuanced discussion about the aging process and a modification of language in curricula that attenuates these negative schemas and, in contrast, stresses the tenets of healthy and normal aging. Finally, future research endeavors employing a variety of study designs are warranted to further our understanding of ageism in clinical settings and assess the utility of educational or clinical interventions in mitigating ageist attitudes in healthcare practice.

## Figures and Tables

**Figure 1 ijerph-19-15870-f001:**
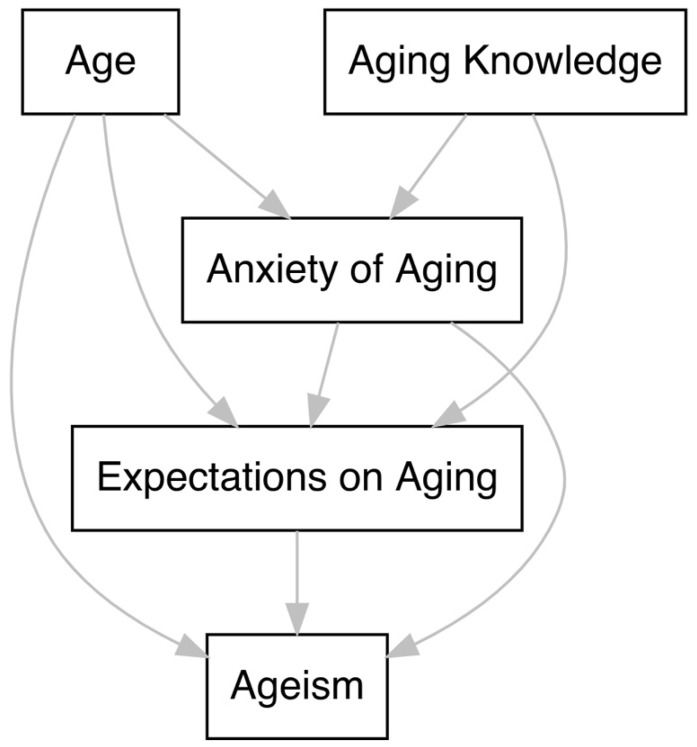
The hypothesized and conceptual path model framework.

**Figure 2 ijerph-19-15870-f002:**
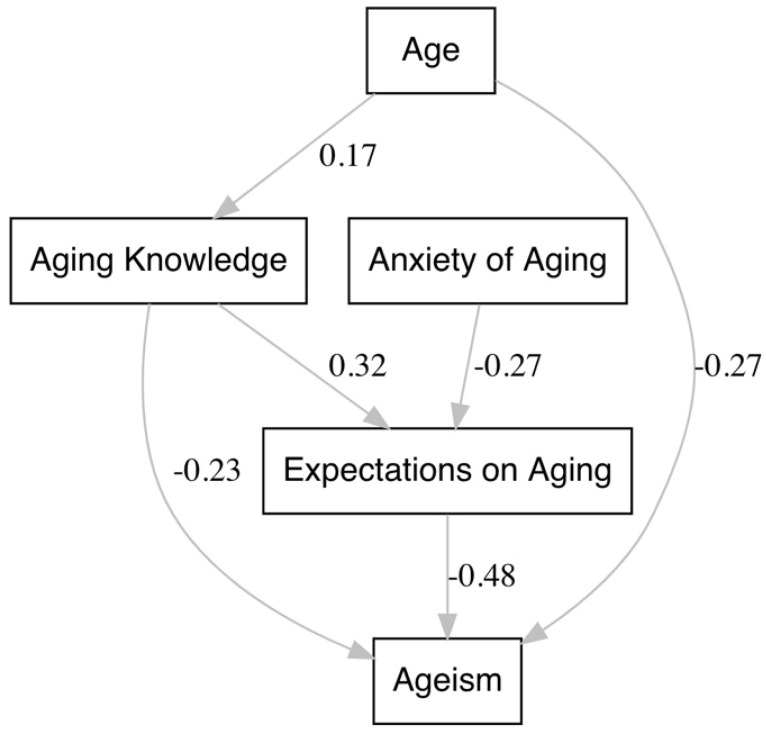
Path plot detailing relevant paths labeled with the standardized path coefficients from the final path analysis model (Model 5). All relationships depicted were significant at the α = 0.05 level.

**Table 1 ijerph-19-15870-t001:** Health Professionals’ Demographic Information (*N* = 140).

Characteristic	Frequency (%) or Mean (SD)
Age	40.3 (13.3)
Gender	
Female	109 (77.9)
Male	31 (22.1)
Highest Education	
Bachelor’s Degree	49 (35)
High School Diploma	14 (10)
Some college, including associate degree	35 (25)
Some post-graduate work or advanced degree	42 (30)
Ethnicity	
Not Spanish, Hispanic, or Latino	116 (82.9)
Spanish, Hispanic, or Latino	24 (17.1)
Race	
American Indian or Alaskan Native	2 (1.4)
Asian	5 (3.6)
Black or African-American	15 (10.7)
Other	5 (3.6)
Selected more than one Race	3 (2.1)
White/Caucasian	110 (78.6)
Marital Status	
Divorced	20 (14.3)
Never married	37 (26.4)
Now married	75 (53.6)
Separated	2 (1.4)

**Table 2 ijerph-19-15870-t002:** Mean scores among health professionals on the measurement tools used in the study.

Characteristic (Max Score)	Mean (SD)	Min–Max
Aging Knowledge Quiz (50)	27.5 (4.6)	16–44
AAS: Total Score (100)	73.7 (8.4)	54–100
Ageism ^a^ (8)	4.3 (2.1)	0–8
Physical Health ERA Score ^b^ (100)	30.3 (15.5)	0–75
Mental Health ERA Score ^b^ (100)	43.2 (19.3)	0–75
Cognitive Health ERA Score ^b^ (100)	31.2 (16.2)	0–75

^a^ Higher score reflects more ageist views; ^b^ Higher scores reflect higher expectations regarding physical, mental, or cognitive health.

**Table 3 ijerph-19-15870-t003:** Relationships amongst the observed study variables using Spearman correlation coefficients.

	1.	2.	3.	4.	5.	6.	7.
1. Age	--						
2. Ageism	−0.30 **	--					
3. ERA-Physical	0.05 *	−0.52 **	--				
4. ERA-Cognitive	0.05 *	−0.42 **	0.66	--			
5. ERA-Mental	0.04 *	−0.63 **	0.68	0.66	--		
6. ERA-Sum ^a^	0.05	−0.59 **	0.87	0.86	0.89	--	
7. AAS	−0.07 **	0.13	−0.21	−0.20 **	−0.17 **	−0.22 **	--
8. FAQ	0.20	−0.47 **	0.33	0.23	0.40	0.38	−0.10 **

** *p* < 0.01; * *p* < 0.05; ^a^ A composite that was calculated by taking the sum of the three ERA subdomains.

**Table 4 ijerph-19-15870-t004:** Goodness-of-fit metrics from all models fit on data.

Model	$\chi^{2}$	df	$\frac{\chi^{2}}{df}$	NFI	CFI	RMSEA	SRMR
Model 1	7.72	1	7.72	0.90	0.90	0.276	0.052
Model 2	7.72	1	7.72	0.90	0.91	0.276	0.052
Model 3	9.32	4	2.33	0.90	0.94	0.113	0.056
Model 4	0.31	3	0.10	1.00	1.00	0.00	0.016
Model 5	0.31	3	0.10	1.00	1.00	0.00	0.016

NFI: Normed Fit Index, CFI: Comparative Fit Index, RMSEA: Root Mean Square of Approximation, SRMR: Standardized Root Mean Square.

**Table 5 ijerph-19-15870-t005:** Standardized coefficients, Z statistics, and *p*-values of direct and indirect paths from the conceptual and final models.

Conceptual	Path	β	Z	*p*
**Model 1**				
*Direct Effects*	Age→Ageism	−0.31	−4.48	<0.01
	EoA→Ageism	−0.56	−9.44	<0.01
	Know→EoA	0.32	3.64	<0.01
	Age→AoA	−0.08	−1.00	0.32
	Know→AoA	−0.05	−0.54	0.59
	Age→EoA	−0.01	−0.17	0.86
	AoA→Ageism	0.00	−0.04	0.96
	AoA→EoA	−0.27	−2.99	<0.01
*Indirect Effects*	Age→EoA→Ageism	0.01	0.17	0.86
	AoA→EoA→Ageism	0.15	2.82	<0.01
**Model 5**				
*Direct Effects*	Age→Ageism	−0.27	−4.03	<0.01
	EoA→Ageism	−0.48	−7.6	<0.01
	Know→EoA	0.32	3.78	<0.01
	AoA→EoA	−0.27	−2.97	<0.01
	Age→Know	0.17	2.49	0.01
	Know→Ageism	−0.23	−2.92	<0.01
*Indirect Effects*	AoA→EoA→Ageism	0.13	2.68	0.01
	Know→EoA→Ageism	−0.15	−3.92	<0.01

Know = Knowledge, AoA = Anxiety of Aging, EoA = Expectations on Aging.

## Data Availability

The data presented in this study are available on request from the corresponding author. The data are not publicly available due to measures to protect participant confidentiality.

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
