# Peer review of "Healthcare Professionals’ Views and Perspectives towards Aging"

_ijerph, 2022, doi:10.3390/ijerph192315870_

Round 1

Reviewer 1 Report

We thank the authors for submitting their manuscript. However, wich require adjustments:

Abstract

- N x n

- The abstract must clearly present the methodology, results (indicating some statistical values) and a synthetic conclusion.

Introduction

- Review punctuation, for example, line 77.

- The introduction seems clear to me and addressing the problem.

Methodology

- It is neccessary to indicate the type of sampling.

Results

- In the correlations it is neccesary to indicate the value of significance (p value) with asterisks.

- The tittles of the tables go to the beginning.

Discussion

- In general, it explains the results well and contrasts them with the literature. However, I suggest the authors rewrite the last paragraph emphasizing the practical implications of their results. How does it help the field of health? What remedies can be developed?

Conclusion

You must respond to the aim based on its results in a more direct way. There is a significant association between...

Author Response

Comment #1: Abstract

- N x n

- The abstract must clearly present the methodology, results (indicating some statistical values) and a synthetic conclusion.

Response #1: We thank the reviewer for this recommendation. We have updated the abstract by adding model estimates and rewriting the concluding sentences to summarize the findings and implications of our work.

Comment #2: Introduction

- Review punctuation, for example, line 77.

- The introduction seems clear to me and addressing the problem.

Response #2:

Thank you for this input, we have reviewed the punctuation and corrected some initial mistakes.

Comment #3: Methodology

- It is necessary to indicate the type of sampling.

Response #3: We thank the reviewer for this suggestion. We have updated the methods section and noted that this was a convenience sample of healthcare workers who participated in the online survey.

We thank the reviewer for this consideration. We remind the reviewer that we detail how healthcare workers were invited to participate in the survey in section “2.1 Setting”. We have added that this was a “convenience sample”. We are happy to include further specific details that the reviewer requests for this section.

Comment #4: Results  

- In the correlations it is neccesary to indicate the value of significance (p value) with asterisks.

- The tittles of the tables go to the beginning.

Response #4: We thank the reviewer for this comment. We have updated Table 3, indicating statistical significance with asterisks. Moreover, we have moved the titles of all tables to the top of the table as opposed to having them on them bottom

Comment #5: Discussion

- In general, it explains the results well and contrasts them with the literature. However, I suggest the authors rewrite the last paragraph emphasizing the practical implications of their results. How does it help the field of health? What remedies can be developed?

Response #5: Thank you for this feedback, we are certainly excited to further expand on the implications of these results and want our suggestions to be clear. We have added to and rewritten portions of the last paragraph in the discussion on page 13-14. We have also expanded on future work, both in terms of future research and policies.

Comment #6: Conclusion

You must respond to the aim based on its results in a more direct way. There is a significant association between...

Response #6: We thank the reviewer for this suggestion. We agree that the conclusions section could be enhanced. We have added some clarifying language succinctly detailing the main associations uncovered in our analysis to the conclusion.

Reviewer 2 Report

I realize that great work and time have been devoted to this paper. It has a lot of strengths, but I think that some changes should be recommended. 

Title: the title does not adequately reflect the content of the paper. Please, try to change it to better inform the readers about the relationships between the variables that you test and also inform them about the quality of your sample.

Abstract:

Less information appears in the abstract. Maybe expanded by adding the most relevant findings. Please, take into account that the abstract is the unique part of your paper that most of the readers could read. Hence, more information would be better. 

Keywords: it is better to enlist your keywords alphabetically. Do not use keywords already captured in the title of the manuscript.

Introduction

The literature revision has some references that are too old. Besides citing some papers from 2001, you can consider some relevant papers on the topic RECENTLY published in other Journals. There are some Journals that suggest a high percentage of references published during the last five years.

Methodology

The Instruments or Questionnaires section needs more information. Please, some examples of items should be provided to the readers. If you can, please inform me about previous studies where the same instrument has been used and the reliability obtained in that research.

Results

Related to your results, I would suggest you clarify for readers which is the content of each figure, for instance, in Figure 2: Are the values the standardized regression coefficients?

Discussion:

First of all, try to better adjust your conclusions to the findings. Or to say in other words, please try to justify more clearly the connection between your conclusions and your findings.

Finally, a section related to limitations, future lines of investigation, and the principal contributions of the research could be attractive. Your paper has a lot of relevant implications for society and policymakers, but you need to elaborate more on this topic.

Author Response

Comment #1: Title: the title does not adequately reflect the content of the paper. Please, try to change it to better inform the readers about the relationships between the variables that you test and also inform them about the quality of your sample.

Response #1:

We thank the reviewer for this meaningful suggestion. We have updated the title of the manuscript. We are open to further suggestions at the discretion of the reviewer.

Comment #2: Less information appears in the abstract. Maybe expanded by adding the most relevant findings. Please, take into account that the abstract is the unique part of your paper that most of the readers could read. Hence, more information would be better. 

Response #2: We thank the reviewer for this comment. We have added more specificity to the abstract by including model estimates and we also altered the conclusions/findings statements.

Comment #3: Keywords: it is better to enlist your keywords alphabetically. Do not use keywords already captured in the title of the manuscript.

Response #3:

Thank you for this input, we have alphabetized the keywords and chosen them such that they are not repetitive with the title.

Comment #4: Introduction, The literature revision has some references that are too old. Besides citing some papers from 2001, you can consider some relevant papers on the topic RECENTLY published in other Journals. There are some Journals that suggest a high percentage of references published during the last five years.

Response #4:

Thank you very much for this feedback, we certainly agree our introduction should include relevant journals. We have gone through our references and added more recent, supporting evidence in conjunction with the older references. Additionally, our discussion section has been updated with a number of new citations, many of which come from within the preceding 5 years. Some older references remain as they do help to highlight that this problem has been well-discussed for the last 20 years.

Comment #5: Methodology, The Instruments or Questionnaires section needs more information. Please, some examples of items should be provided to the readers. If you can, please inform me about previous studies where the same instrument has been used and the reliability obtained in that research.

Response #5:

Thank you for this suggestion, we have added examples of tools to sections 2.5, 2.6, 2.7 and 2.8 on page 4. We have also updated the language to indicate that each tool is validated, the validation studies were previously cited. The exception is the WHO ageism scale, which is not validated. This is discussed in the limitations paragraph as well as the conclusions sections, where we point to a need for the development for a robust scale that measures the multidimensional nature of ageism. We also cite other relevant research that has proposed a similar call to action. 

Comment #6: Results, Related to your results, I would suggest you clarify for readers which is the content of each figure, for instance, in Figure 2: Are the values the standardized regression coefficients?

Response #6: We thank the reviewer for this suggestion. We remind the reviewer that the original figure caption already specified that the figure provided “standardized path coefficients”. Nevertheless, we have updated the language in the captions for Figure 1 and Figure 2 to make them more clear to readers. We are happy to incorporate any other suggestions made by the reviewer to make it easier to understand.

Comment #7: Discussion, First of all, try to better adjust your conclusions to the findings. Or to say in other words, please try to justify more clearly the connection between your conclusions and your findings.

Response #7: Thank you for this feedback, we have added a quick summary of our findings to the conclusion in order to make the connection clearer, as well as some transition statements about how we drew our conclusions. We have also edited the end of the discussion to reflect this. We have also expanded the conclusions section significantly to address this.

Comment #8: Finally, a section related to limitations, future lines of investigation, and the principal contributions of the research could be attractive. Your paper has a lot of relevant implications for society and policymakers, but you need to elaborate more on this topic.

Response #8: We thank the reviewer for this very meaningful suggestion. We have expanded the commentary on the policy implications of our work (502-514). We also have a very detailed section for our limitations (page 13, lines 461-480). Our conclusions section has been revamped significantly with expanded commentary on implications and future directions.

Round 2

Reviewer 1 Report

Thank you for the adjustments!!!

Author Response

We thank the reviewer for their kind comment and for taking the time to review our manuscript